# Stocking Density Affects Stress and Anxious Behavior in the Laying Hen Chick During Rearing

**DOI:** 10.3390/ani9020053

**Published:** 2019-02-10

**Authors:** Kaya von Eugen, Rebecca E. Nordquist, Elly Zeinstra, Franz Josef van der Staay

**Affiliations:** 1Behavior and Welfare Group, Department of Farm Animal Health, Faculty of Veterinary Medicine, Utrecht University, P.O. Box 80151, 3508 TD Utrecht, The Netherlands; r.e.nordquist1@uu.nl (R.E.N.); e.c.zeinstra@uu.nl (E.Z.); F.J.vanderStaay@uu.nl (F.J.v.d.S.); 2Biopsychology, Institute of Cognitive Neuroscience, Faculty of Psychology, Ruhr-University Bochum, 44780 Bochum, Germany

**Keywords:** poultry, rearing phase, stocking density, crowding, stress, corticosterone, anxiety

## Abstract

**Simple Summary:**

‘Crowding’, keeping too many birds per m^2^, is one of the largest welfare concerns in the poultry industry. It is therefore worrisome that there is a gap in research investigating the effects of high stocking densities during the rearing phase of laying hens. This study evaluated anxious behavior and corticosterone levels, a hormone involved in the stress response, during the first 10 weeks of laying hen chicks housed under three different crowding conditions: undercrowding, conventional crowding, and overcrowding. We found that overcrowded chicks displayed more anxious behavior compared to undercrowded chicks. Corticosterone levels were elevated in both extreme groups in week 3, but dropped to values of the conventional crowding group at week 10. We conclude that current conventional stocking densities do not seem to impair the welfare state of the laying hen chick, and that a three-fold in- or decrease of density influences stress and anxiety, but within the adaptive capacity of the chick. Important side-notes to this conclusion are that an increase of stocking density did result in a slower rate of adaptation, and that we currently do not know if there are long-term consequences of different crowding densities reaching into the laying period.

**Abstract:**

The recent increases in stocking density, in extreme cases resulting in ‘crowding’, have a major impact on poultry welfare. In contrast to available research on adult laying hens, there is a gap in the literature studying the rearing phase. The present study investigated the effects of stocking density during the rearing period on the welfare of the laying hen chick. The chicks were housed under one of three crowding conditions, increasing with age: undercrowding (500-1000-1429 cm^2^ per chick), conventional crowding (167-333-500 cm^2^ per chick), or overcrowding (56-111-167 cm^2^ per chick). The parameters evaluated encompassed behavioral and physiological factors related to anxiety and stress. We found that during the first 6 weeks, overcrowded chicks displayed more anxious behavior than undercrowded chicks, and both extreme densities induced higher corticosterone levels compared to chicks housed under conventional crowding. At 10 weeks of age, plasma corticosterone had dropped to the level of conventional crowding group in both groups, whereas feather corticosterone remained high only in the overcrowded group. We conclude that current conventional stocking densities do not seem to impair the welfare state of the laying hen chick, and that a three-fold increase or decrease of density influences corticosterone levels and anxious behavior, but within the adaptive capacity of the chick. Important side notes to this conclusion are that an increase of stocking density did result in a slower rate of adaptation, and that there could be long-term consequences of both the different stocking densities and/or increased costs of adaptation.

## 1. Introduction

Since the industrial revolution, poultry farming production systems have been scaling up to increase economic returns. An important consequence is an increase of stocking density in both the rearing and laying phase of layer hens (*Gallus gallus domesticus*). If taken to the extreme, this can lead to ‘overcrowding’, keeping too many birds per square meter, which can have a major impact on welfare. First, it may cause discomfort, distress, and restrictions on behavior, and second, it may limit feed or water intake and lead to injury or disease [1]. Additionally, it has been shown to be an important predictor for the development of feather pecking [2,3], which is one of the most significant challenges in poultry welfare [4]. This is not surprising, since in general it is well known that environment and experience during early life can have long-term effects [5,6].

It is therefore worrisome that there is a gap in research on the effects of stocking densities during the rearing phase, see Figure 1, of the laying hen. In contrast to the abundance of literature on the effects of crowding on welfare in broilers (for a review see [7]) and adult laying hens [5], there are only approximately a dozen papers focusing on laying hen chicks (Table 1), and several concerns can be raised on the available literature. For one, most of the research was executed over a decade ago. This is problematic since both breeding lines using conditions have changed over time, making it difficult to apply the results to the current situation. Moreover, the research had a strong focus on development and performance (i.e., the production of eggs), and parameters such as normal behavior and welfare were rarely discussed; it appears these themes have entered the scope of research only more recently. These factors combined complicate forming an informed consensus on the impact of stocking density during rearing on laying hen chicks and adults, specifically regarding their welfare.

The concept of animal welfare can be defined and assessed in a multitude of ways. Here we approach it as a dynamic state level that revolves around the ability for the animal to adequately adapt to the aversive intrinsic and extrinsic factors to achieve a state perceived as positive [8]. It is difficult to assess the perception of a state in animals directly; thus, in this study we will approximate the multi-faceted concept of welfare by focusing on the closely interconnected phenomena stress and anxiety.

Stress is a state in which (extreme) extrinsic and intrinsic demands push beyond an organisms’ natural regulatory capacity [9]. Crowding can be seen as such an extreme environmental demand [1]. Physiologically, stress can be expressed in levels of the hormone corticosterone (CORT). It is released by the hypothalamic-pituitary-adrenal (HPA) axis in response to both appetitive and aversive stimuli, and acts upon behavior, metabolism, and the immune system [10]. Chronic high levels of CORT may result in growth retardation, impaired immunocompetence, and disrupted species-specific behavior [11]. Until now, no studies have evaluated CORT levels in young birds subjected to different stocking densities. However, a positive correlation between crowding and the physiological stress response has been demonstrated by the ratio of heterophils and lymphocytes [12], which is indicative of plasma CORT levels [13].

Since corticosterone has been implied in such a wide variety of physiological processes, its usefulness in the assessment of stress and welfare has been questioned [14]. It is therefore important to combine hormone values with behavioral measures that are informative of stress. On the behavioral level, stress is closely related to anxiety. Anxiety can be defined as general distress and is characterized as a response to a potential danger. It is a state that ranges from low to high and is reflected in the level of fearfulness and sociality [15]. Fearfulness is seen as the immediate behavioral response to a stimulus or event perceived as a threat, and it co-occurs with an immediate rise in CORT levels [16]. Similarly, domestic fowl that receive CORT via an infusion pump display more fearful behavior than fowl with a sham pump [13]. Fearful behavior can be measured in an open field (OF) test, where general activity has an inverse relationship with fearfulness [17,18,19]. Importantly, the intensity of the fearful response in the OF can be considered a measurement of overall anxiety, because the OF does not impose a direct threat on the chick [20].

Sociality, the need to be with conspecifics, increases with higher levels of anxiety and stress. Marin et al. [21] employed a runway test with a stimulus bird in the goal box and found that chicks subjected to an acute stressor displayed a shorter latency to leave the start box and spent more time near the stimulus bird. They interpret this as anxious chicks demonstrating increased motivation for social reinstatement. Sociality can be assessed in a Y-maze test, which differentiates between social motivation versus exploration and foraging motivation [22,23]. General activity in the first trial of the Y-maze can be interpreted in a similar way as activity in the OF; again, a novelty is imposed on the chicks, and the intensity of the behavioral response can be interpreted as a measure of anxiety. Additionally, the rate of change in behavior over the trials is also indicative of anxiety, since it reflects the process of familiarization or adaptation to the test environment [24]. Furthermore, underlying sociality is measured by the tendency to remain in close proximity to the social stimulus, which is also known as the ‘social reinstatement response’ [25], instead of moving away from the social stimulus to forage.

The present study aimed to elucidate the consequences of early life crowding on welfare by evaluating stress, fear, and sociality in Brown Nick chicks housed under three different crowding conditions: overcrowding (OC), conventional crowding (CC), and undercrowding (UC). The experiment took place in small groups in a laboratory set-up, where besides the stocking density all environmental variables were strictly controlled and kept equal across the groups. The aim was not to copy the exact conditions of housing in the poultry industry, but to specifically isolate the effects of crowding on physiology and behavior, and ultimately welfare. We evaluated resting stress levels by assessing circulating plasma-CORT [26], and CORT deposition in feathers for an assessment of (putative) stress effects over a longer timeframe [27,28,29]. Fearfulness was measured in an open field (OF), and sociality in an Y-maze test. Lastly, body weight was assessed as a gross indicator of growth. We believe that the combination of these parameters, and the mentioned interconnectivity between them, will yield a more complete indication of the effects of stocking density during the rearing period of the laying hen on ultimately welfare.

## 2. Materials and Methods

### 2.1. Welfare Note

This study was reviewed and approved by the local animal ethics committee (DEC; Dier Experimenten Commissie, approval number 2013.l.11.085) of Utrecht University, the Netherlands. This committee has been issued a license by the governmental Central Animals Experiments Committee in accordance with the recommendations of the EU directive 86/609/EEC. All effort was taken to minimize the number of animals used and their suffering.

### 2.2. Animals

Forty-two female Brown Nick chicks were obtained on the day of hatching from Verbeek Hatchery and Rearing (Lunteren, The Netherlands). The chicks were not beak-trimmed or vaccinated. All chicks were housed in the same room under similar conditions, except for the ‘crowding-factor’: undercrowding (UC), conventional crowding (CC), and overcrowding (OC), see Table 2. On the day of arrival, each chick was randomly assigned to one of these three conditions, resulting in groups of 14 chicks per cage and thus crowding density.

On day 2, the chicks were marked individually with either green or purple pig paint (Kruuse, Denmark) in different combinations. In addition, the chicks were ringed for individual identification in week 2. From day 1, all the chickens were provided with starter chick food (Besterfood Opfokmeel, The Netherlands) and water ad libitum. From week 3 onwards, enrichment was provided by scattering mixed grain on the floor of the cages once a day. During the first week, the chicks were weighed every day, and after week 1 they were weighed twice a week to monitor growth. All chickens were euthanized by cervical dislocation when 10 weeks old, see Figure 1.

### 2.3. Housing

All cages were square-shaped and consisted of two walls of medium density fiber (MDF), and two walls of chicken fence on a wooden frame. All cages were 60 cm high. The floor was covered with wood shavings. Every cage contained a feeding trough (length 30 cm) and a drink silo (week 0–3: diameter 16.5 cm, week 4–10: diameter 22 cm), placed in a similar composition in each cage.

The three cages were placed in a temperature-controlled (18–22 °C) and well-ventilated room. Lights were on from 07:00 a.m. to 07:00 p.m. Chicks of different cages were unable to see each other because every open wall faced a closed wall of a neighboring cage. During the first three weeks every cage was provided with a heat lamp for sufficient warmth, and with a min/max thermometer to monitor temperature. After three weeks the cages were closed off with mesh wire to prevent chicks from flying out of their pen.

The crowding factor was based on numbers used in the industry (personal communication with the two largest layer hen rearing facilities in the Netherlands) to model conventional crowding, and a factor of three smaller or bigger to design over- and undercrowding conditions respectively. Similar to industry-practice, the chicks’ housing was enlarged in week 4 and week 7 by means of an enlargeable mechanism inherent to the cages, ensuring that the chickens did not have to move but could stay in their familiar home-pen, see Table 2.

### 2.4. Behavioral Tests

The chicks were tested in an open field when 21–22 days (three weeks) old, and in a social versus foraging Y-maze when 56–60 days old (eight weeks), see Figure 1. Order of testing was randomized by consecutively testing the chicks according to their randomly assigned number (1–14) and alternating between the groups during testing.

#### 2.4.1. Open Field Test

##### Apparatus

The open field apparatus (Figure 2, left panel) measured 150 × 150 cm (approx. 2.25 m^2^). The 65 cm high walls were made of grey Polyvinyl chloride (PVC), fixed by connectors of stainless steel. The floor was divided with white chalk lines in a 3 × 3 checked pattern, where each square measured 50 × 50 cm. A camera was mounted above the observation pen, allowing observations from a monitor situated in the testing room but out of view for the chick being tested. This allowed scoring of behavior without disturbing the chick during the test. Behaviors were scored live using the observation software JWatcher 1.0 [44].

##### Procedure

A chick was gently caught from its home pen and placed in the center of the open field. The observation time of 10 min started as soon as the experimenter was out of sight of the chick in the open field. We scored: latency to leave the starting square, number of lines crossed with both feet, total time spent walking and number of distress calls (high-pitched peep of high energy). Immediately after the observation period, the chick was placed back in its home pen. All chicks were tested between 9:00 a.m. and 5:00 p.m. on two consecutive days.

#### 2.4.2. Y-Maze Test Social Versus Foraging

##### Apparatus

The Y-maze apparatus (Figure 2, right panel) consisted of one long arm (150 × 50 cm) and two shorter arms (125 × 50 cm). The 65 cm high walls were made of grey PVC fixed by connectors of stainless steel. At the beginning of the long arm a compartment of 50 × 50 cm served as start box. A guillotine door provided access to the Y-maze. At the end of the shorter arms, a compartment (size 50 × 50 cm) held either the social stimulus or the foraging stimulus. The foraging stimulus consisted of mixed grain dispersed in mixed wood shavings and was continuously accessible to the test chick. The social stimulus consisted of a familiar chick from the home pen, separated from the rest of the Y-maze apparatus with a fine wire mesh. The apparatus was divided in areas with white chalk lines on the floor for proper analysis: start box, zone 1, midzone, social arm, and foraging arm. A camera was situated above the apparatus in order to score the behavior live from a connected monitor.

##### Procedure

After both stimuli were prepared, a chick was gently caught from its home pen and placed in the start box. The 10-min test started by opening the sliding door between start box and maze. We scored: latency to leave the start box, time spent in each zone, and the number of lines crossed. All behaviors were scored live using JWatcher 1.0. After 10 min both the social stimulus chicken and the test chicken were gently caught and placed back in their home pen. Each chick completed four trials on four consecutive days. To control for the possible effect of an arm preference, stimuli were systematically alternated between end compartments over the four trials. Whether a chick would start with the social stimulus in the left or right arm alternated per chick and was equally divided per group. Moreover, to prevent an influence from a social preference or established hierarchy, two chicks from their home pen were selected randomly as the fixed social stimuli and alternated over the four trials for each test subject. These chicks were excluded from the Y-maze test.

### 2.5. Glucocorticoids

#### 2.5.1. Plasma

At 6 and 10 weeks of age blood-samples were obtained from all chicks, see Figure 1. Samples were taken between 11:15 a.m. and 12:45 p.m. After a chick was carefully caught from its home pen, a blood sample was taken by an experienced poultry veterinarian within 3 min with either a short 21G × 5/8-inch needle (Terumo Neolus, Belgium) and 3 mL syringe (Braun Omnifix, USA), or a size 11 scalpel blade (Swann-Morton, England) from the V. Ulnaris. Approximately 2 mL blood was collected in a 2 mL EDTA tube (BD Vacutainer, USA) and stored in a cooler. Immediately thereafter, the samples were centrifuged for 10 min at 20 °C at 2000G (B. Braun Sigma 4K10, USA). From the plasma, 300–1000 µL was pipetted in a smaller container and stored at −20 °C until analysis of all samples (corticosterone ELISA LDN GmbH & Co, Germany).

#### 2.5.2. Feathers

We collected the primary feathers 2 and 8 (see Figure 3A) from both the left and right wing in week 3 and 10, see Figure 1. These feathers were chosen since they are used most in the available literature [25], hence allowing for comparison. The complete feather was carefully pulled and stored in an airtight plastic bag. Primary 2 and 8 were stored separately but left and right were grouped together. The feathers were stored in a dark compartment at room temperature until further analysis.

The extraction of corticosterone (CORT) from the feathers relies on an 80% methanol (Merck 1.06009, Germany) extraction technique and is based on a combination of the method of Bortolotti et al. [27] and Berkvens [28]. The feather was cleaned with 100% methanol using an aerosol spray, dried under the fume hood, weighed (Scout Pro SPU 123, Ohaus Corporation, USA), and measured. Next, the calamus, rachis, downy barbs, and tip of the feather were removed, and the remaining vanes were sectioned with scissors into flakes of <3 mm^2^ (see Figure 3B). The flakes were collected in a 15 mL Falcon tube and stored at −20 °C. For the actual extraction, a basis of 5 mL 80% methanol was added, plus 1 mL 80% methanol per 0.03 g flakes, followed by incubation overnight on a roller bank (Stuart SRT9D, Bibby Scientific, England), speed 30, at room temperature. The solutions were centrifuged for 10 min at 2200G (B. Braun Sigma 4K10, USA) and 4 mL supernatant was pipetted in four 2 mL Eppendorf tubes and stored at −20 °C. Lastly, the methanol extracts were dried in a vacuum pump (Savant Speedvac AES 1000-240, Thermo/Fisher Scientific, Germany) on medium heat and stored at −20 °C until analysis.

CORT concentration was determined in triplicate with a corticosterone enzyme immunoassay kit (number 500655-S1, Cayman Chemicals, BioConnect, The Netherlands). From each sample, two of the four Eppendorf tubes with reconstituted feather extracts were combined for analysis. All samples were measured in eight separate assays performed on four consecutive days. Every assay contained samples from each of the three experimental groups, systematically divided over the assay. Feather CORT values are expressed in milligram per gram vanes.

### 2.6. Staistical Analysis

All statistics were calculated with SAS Studio 3.4 (Basic Edition), running on a Linux system within a Virtual Machine environment on a MacBook Pro. The Shapiro-Wilk test was used to assess normality (SAS UNIVERIATE procedure), homogeneity of covariance matrices was evaluated with a Box M test (SAS DISCRIM procedure), the assumption of sphericity was checked using a Mauchly’s test (SAS GLM procedure), and homogeneity of variances was assessed with the Levine’s test (SAS GLM procedure)

Body weight was measured on 27 different time-points. Since Mauchly’s test indicated that the assumption of sphericity had been violated, we assessed the separate ANOVA with Greenhouse–Geisser correction. From the OF we considered latency to leave (in s), the number of lines crossed, time spent walking, and the number of distress calls. During the OF, two chicks of UC, two chicks of CC, and one chick of OC escaped from the test pen during the test, thus their data were excluded. Latency to leave and number of lines crossed violated the assumption of normality, therefore a log10 and square root transformation was applied respectively to meet this assumption. In the Y-maze tests we assessed the parameters latency to leave (in s), arm preference and number of lines crossed over four consecutive sessions. Arm preference was a compounded variable calculated from time spent (in s) in the foraging arm subtracted from time spent in the social arm. Thus, a value of zero indicated no preference, a positive value indicated a preference for the social arm, whereas a negative value indicated a preference for the foraging arm. The variables latency to leave the start box and number of lines crossed violated the assumption of normality; therefore, they were log10 and square root transformed respectively and normality was no longer violated. Since the assumption of sphericity was violated (Mauchly’s test) in latency to leave with ε < 0.7, we looked at the MANOVA score for this parameter. For arm preference and lines crossed, Mauchly’s test was significant, but ε was > 0.7, thus, the separate ANOVAs were considered with Greenhouse-Geisser correction. Feather glucocorticoid deposition was analyzed for feathers 2 and 8 separately, since there is no relation between the separate feathers in terms of CORT deposition [27], and since the variables violated the assumption of normality, the data were log10 transformed to meet this assumption.

For all dependent variables, the effects of the crowding conditions were analyzed using a parametric (multivariate) analysis of variance (MANOVA Pillai’s trace; ANOVA; PROC GLM procedure). When a significant effect of crowding factor was found, a post hoc analysis for pairwise comparisons between the three crowding conditions was executed with a Tukey HSD test (SAS GLM MEANS procedure). To highlight differences of within-subject measurements and interaction effects (for body weight, Y-maze test behaviors, and glucocorticoid levels in plasma and feathers), a repeated measures analysis was included (SAS GLM REPEATED procedure), with a contrast analysis (PRINTE function) for pairwise comparison within-subjects. The false discovery rate (FDR; SAS MULTTEST procedure) was used to correct for inflation of the α-level in multiple comparisons. In all statistical tests, an effect was considered significant at *p* < 0.05. Effect sizes eta (η) were estimated (SAS GLM procedure, α = 0.10) with η2. Values of 0.01, 0.06, and 0.14 were considered as small, moderate and large effects, respectively [45].

## 3. Results

### 3.1. Body Weight

There was no difference in body weight between the groups across all time-points, F(52,936) = 1.57, *p* = 0.212.

### 3.2. Open Field Behaviors

There was an overall effect of crowding condition on behavior in the open field F(10,62) = 3.27, *p* = 0.002. The groups differed in latency to leave the starting field, F(2,34) = 4.41, *p* = 0.025, η^2^ = 0.21 (Figure 4A); time spent walking, F(2,34) = 9.19, *p* = 0.003, η^2^ = 0.35 (Figure 4B); number of lines crossed, F(2,34) = 7.00, *p* = 0.007, η2 = 0.29 (Figure 4C); and number of distress calls omitted, F(2,34) = 4.74, *p* = 0.025, η^2^ = 0.22, (Figure 4D). Post hoc analysis showed that chicks from the UC group displayed a shorter latency to leave than chicks from the CC group. Moreover, the UC group crossed more lines, spent more time walking and emitted more peeps than the OC group.

### 3.3. Y-Maze Behaviors

The overall analysis showed that there was an effect of the crowding conditions on behavior in the Y-maze, F(2,33) = 4.42, *p* = 0.020. There was no effect of crowding conditions on latency to leave over the four trials combined, F(2,33) = 1.96, *p* = 0.158, and no interaction effect between crowding conditions and trials, F(6,64) = 1.12, *p* = 0.362. Latency to leave differed between trials, F(3,31) = 18.66, *p* < 0.001 (Figure 5A). The contrast analysis showed that latency to leave in trial 1 was longer than in trial 2, F(1,33) = 34.80, *p* < 0.001, trial 3, F(1,33) = 56.49, *p* < 0.001, and trial 4, F(1,33) = 53.47, *p* < 0.001. Latency to leave in trial 2 was longer than in trial 3, F(1,33) = 13.79, *p* = 0.0010, and trial 4, F(1,33) = 15.54, *p* < 0.001.

Arm preference was affected by crowding conditions, F(2,33) = 4.42, *p* = 0.020, and trial number, F(3,99) = 6.11, *p* = 0.002 (Figure 4B). Follow up analysis revealed a preference only in trial 2, F(2,33) = 4.11, *p* = 0.029, η^2^ = 0.26, and for trial 1, F(2,33) = 4.11, *p* = 0.051, η^2^ = 0.20, which only approached significance but showed a large effect size. Post hoc tests showed that during trial 2, chicks from the UC group spent less time in social contact compared to the chicks of both the CC and the OC group. In trial 1, the UC chicks displayed less social behavior only compared to the chicks from the OC group. Contrast analysis showed that, all chicks combined, they spent less time in social contact in trial 1 compared to trial 2, F(1,33) = 8.85, *p* = 0.012; and trial 4, F(1,33) = 9.32, *p* = 0.012. This also holds for trial 3 compared to trial 4, F(1,33) = 8.64, *p* = 0.012.

The groups differed in the number of lines crossed over the four trials combined, F(2,33) = 4.43, *p* = 0.020, and this behavior changed across trials, F(3,99) = 3.81, *p* = 0.017 (Figure 4C). The separate ANOVA’s per trial revealed that the group differed for number of lines crossed only in trial 1, F(2,33) = 4.99, *p* = 0.026, η^2^ = 0.23; and trial 2, F(2,33) = 5.18, *p* = 0.026, η^2^ = 0.24. Post hoc comparisons revealed that for both sessions the chicks from UC and CC crossed more lines than the chicks from OC. Analysis of all chicks combined between trials showed that they crossed more lines in trial 2 compared to trial 1, F(1,33) = 14.80, *p* = 0.003; and trial 4, F(1,33) = 7.96, *p* = 0.024.

### 3.4. Plasma CORT

The analysis of the plasma CORT levels on two time points revealed an overall effect of the crowding conditions, F(2,39) = 4.13, *p* = 0.024, age, F(1,39) = 22.95, *p* < 0.001, and their interaction, F(2,39) = 6.73, *p* = 0.003 (Figure 6A). Separate ANOVA’s per time-point showed that plasma CORT differed between the groups in week 6, F(2,39) = 8.10, *p* = 0.002, η^2^ = 0.29, but not in week 10, F(2,39) = 0.13, *p* = 0.88, η^2^ = 0.0067. Post hoc comparisons revealed that in week 6 both the chicks from UC and OC had higher values compared to the chicks from CC.

### 3.5. Feather CORT

In feather 2, there was an effect of crowding condition, F(2,39) = 27.67, *p* < 0.001, age, F(1,39) = 64.49, *p* < 0.001, and their interaction, F(2,39) = 5.70, *p* = 0.007, on CORT levels (Figure 6B). Groups differed from each other in week 3, F(2,39) = 10.37, *p* < 0.001, η^2^ = 0.35; and in week 10, F(2,39) = 24.37, *p* < 0.001, η^2^ = 0.56. Post hoc analysis revealed that in week 3 feather 2 of chicks from UC and OC contained higher levels of CORT than the chicks from CC. In week 10, the chicks from OC displayed higher levels of CORT in feather 2 compared to the chicks from both CC and UC.

In feather 8, crowding conditions, F(2,38) = 12.68, *p* < 0.001, as well as age, F(1,39) = 162.43, *p* < 0.001, influenced CORT levels (Figure 6C). Moreover, CORT levels varied between the two time-points differently between the groups, F(2,38) = 13.14, *p* < 0.001. Separate ANOVAs per time-point revealed that there was no difference between the groups in week 3. In week 10 the groups did differ, F(2,38) = 20.27, *p* < 0.001, η^2^ = 0.52. The post hoc analysis revealed that feather 8 of chicks from OC displayed higher CORT values than those of UC and CC chicks.

## 4. Discussion

The present study investigated behavioral and physiological measures of stress and anxiety during the rearing period to approximate the welfare state of the laying hen chick raised in different stocking densities. The chicks were housed under one of three crowding conditions: undercrowding (UC), conventional crowding (CC), or overcrowding (OC). We evaluated fearfulness in the open field (OF) test, sociality in the Y-maze test, plasma values for the resting levels of corticosterone, and feather deposition as a measure of long-term corticosterone values. We found that undercrowded chicks were more active and needed less social reinstatement compared to the overcrowded group. Both extreme crowding conditions induced higher CORT levels compared to chicks housed under conventional crowding during the first six weeks of the experiment. At ten weeks, circulating plasma CORT had dropped to CC levels in both the UC and OC groups. However, feather CORT deposition in the overcrowded group remained high. Below we will discuss the interplay between CORT values and general activity levels, and how we should address our findings in relation to the chicks’ welfare.

### 4.1. Body Weight

Most of the previous studies reported a lower body weight as a consequence of higher stocking densities [30,32,33,40], which contrasts with this study where we found no weight differences between the groups. Possibly, differences in body weight only become apparent later in the rearing period, as has been suggested by Bell [46]. This was corroborated by Anderson & Adams [30], who reported body weight differences only from 12 weeks of age onwards. In this study, chicks were sacrificed at the age of 10 weeks, possibly before weight differences became apparent. Future studies should prolong investigation time and extend into the laying period to elucidate these effects.

### 4.2. Stress and Anxiety

The main aim of this study was to assess the impact of crowding density on welfare by looking at stress and anxiety. We found that chicks reared under high crowding densities were less active and displayed a higher need for social reinstatement. These findings indicate that OC chicks experienced more anxiety when faced with novelty and separated from their pen mates. The higher levels of anxious behavior in OC are in line with increased CORT levels in feathers and plasma at three weeks and six weeks of age respectively, compared to chicks reared under lower densities. This apparent relationship between CORT levels and anxious behavior corroborates previous findings. Namely, Jones et al. [47] classified Japanese quail as either low stress (LS) or high stress (HS) based on plasma CORT levels following a restraint and compared levels of fearful behavior in an OF. They demonstrated that LS quails display less freezing and shorter call latencies than HS chicks and concluded that there is a positive correlation between anxiety and CORT activation. Satterlee & Marin [48] furthermore demonstrated that exposing quails to a stressor before an OF induced more freezing and immobility compared to non-stressed individuals.

Interestingly, chicks reared under the lowest density (UC) show a different pattern. Behaviorally, they were most active and demonstrated little need for social reinstatement. However, this is in contrast with CORT findings since levels of CORT in feathers and plasma during the first six weeks of the experiment in UC chicks were similar to those of the chicks reared under the highest density. We propose that the deviating behavioral data are a consequence of the experimental design used; the UC chickens were already well habituated to larger spaces and social isolation. When Hall [17] first described the open field test for rodents, it had to meet two criteria: (1) the tested animal should be unfamiliar with the test arena, and (2) the test arena should be larger than the home cage. Since the dimensions of the open field apparatus were the same for each group, while the cages of the three crowding conditions differed in size, the experience of space may have been very different for each of the three groups. Since one of the main fearful aspects of the OF is its size, it is highly probable that the rearing conditions influenced fear levels through familiarization with space. Jones [49] investigated the consequences of repeated exposure to the OF and demonstrated that, over successive trials, the chicks displayed less freezing behavior, walked more, and emitted more peeping. It appears that something similar occurred for the chickens reared in the UC condition and their perception of the OF.

In this case, we do make the assumption that if provided with more space in their home pen, the chicks will also make use of it and hence become familiar with large spaces and maintain higher inter-individual distances. This has not been demonstrated in laying hens, but Leone & Estevez [50] showed that the movement pattern of young broiler chicks was primarily determined by the size of their home pen. In other words, when provided with space, the chickens make use of it.

An alternative explanation for the increased CORT levels in the undercrowded group is that the higher values are a consequence of increased physical activity, since a main effect of glucocorticoids is to mobilize energy reserves to meet increased metabolic demands [51]. Indeed, in response to exercise, sheep have increased plasma cortisol levels [52]. However, whereas it might be of crucial importance on the short-term to anticipate energetic needs, prolonged higher levels of glucocorticoids can have detrimental effects [10]. In this study, the undercrowded group showed increased CORT levels for at least the first six weeks, and we currently do not know what the consequences could have been for adult laying hens raised under these circumstances. Perhaps, and this is purely speculative, the increased physical demands would negatively affect final body weight, laying performance or even mortality rate.

To conclude, the UC chickens were possibly too familiarized with the large free space of the OF to show a fear response, and it remains to be demonstrated whether a heightened fearful response in UC chicks might have become apparent in a modified test (e.g., a much larger open field apparatus). Alternatively, the increased CORT levels are a consequence of higher activity levels in this group.

### 4.3. Welfare

Finally, it appears that chickens reared under conventional crowding did not have high levels of stress or anxiety at any time-point in this experiment. In other words, the chicks reared under conventional crowding conditions appeared to be unimpaired in their welfare compared with chicks in the other two crowding conditions. Namely, the present study suggests that chicks housed under lower and higher crowding conditions were impaired in their welfare during the first six weeks of the experiment as indicated by higher levels of CORT and anxious behavior. At this time-point, both more extreme housing densities might be experienced as aversive to chicks of this strain.

However, the presence of stress and anxiety are in itself not indicative of an impaired welfare state, since there are numerous examples of the beneficial aspects of these emotional states [14,15]. As outlined before, the crux of a positive welfare state is the ability of the animal to adapt to the event or situation it perceives as negative. Only following prolonged exposure incapacitating this adaptability, in this study indicated by chronic stress and/or pathological anxiety, we would speak of an impaired welfare state [8].

In fact, in week 10, both undercrowded and overcrowded chicks display activity and CORT levels equal to CC chickens; both groups were apparently able to adapt to their housing situation. Most probably, over the years, Brown Nick laying hens have been selected in concordance with their performance in the provided environment in order to increase economic returns [53]. Possibly, through a long history of poultry farming practices, a suitable stocking density has been established as a result of experience (not as a result of systematic scientific research). Increasing or decreasing the stocking density during the rearing phase does affect the welfare state of the chicks on a short term, but this strain is capable and able to adapt to these deviating environmental demands. Important to note is that it does appear that OC chickens displayed a lower rate of adaptation, since CORT deposition in their feathers was still high in week 10. Thus, a high stocking density may be more demanding than a low stocking density.

An important side-note to this conclusion is that our study is inconclusive about the long-term consequences of these different crowding conditions, and as of now we do not know how behavior, stress response, and breeding performance will develop in the adult laying hens. It is imaginable that the increased demands of the process of adaptation for the undercrowded and overcrowded group have long-term effects in terms of growth and physiology [54]. However, these results from the rearing phase are in line with a prevailing conception that one of the most important aspects of good welfare is “a close match between early-life environment and adult environment” [6] (p. 92). This touches upon the capacity to adapt and thereby secure a minimum level of welfare, which all chicks in this study eventually attained.

## 5. Conclusions

This study demonstrates that the currently employed stocking density in the Netherlands during the rearing phase appears to not impair the welfare state of the laying hen chick. Interestingly, a threefold increase or decrease of the crowding density induces more anxious behavior and higher corticosterone levels on the short term, but these drop to levels of conventional crowding chicks towards the end of the experiment. This can be interpreted as that this type and level of environmental challenge is within their adaptive capacity. A crucial side-note to this conclusion is that we currently do not know what the long-term consequences are of either the direct impact of the crowding densities or the costs of adaptation.

## Figures and Tables

**Figure 1 animals-09-00053-f001:**
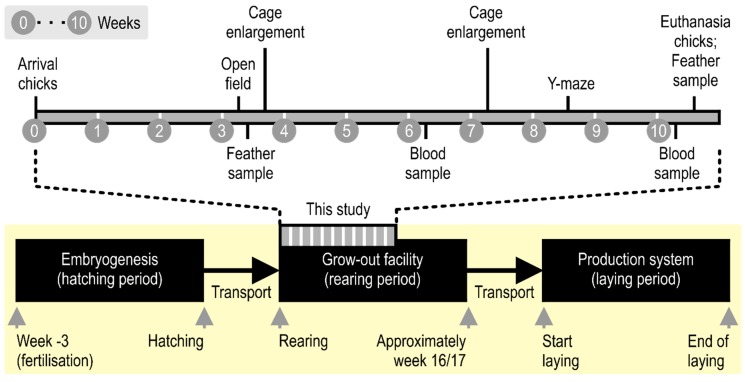
The production of the laying hen consists of three phases that take place in different facilities. After embryogenesis, one-day-old chicks (after hatching) are moved to the grow-out facility for the rearing period. Here they are kept until approximately week 16/17, when they start laying eggs, and are moved to the production system for the laying period. This study addresses the first 10 weeks of the rearing phase.

**Figure 2 animals-09-00053-f002:**
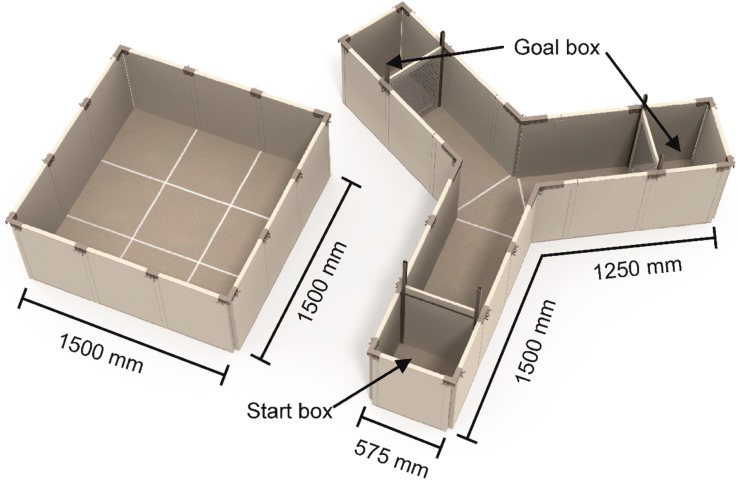
Open field (left panel) and the Y-maze (right panel). Chicks were placed in the center of the OF, and behavior was monitored for 10 min. The Y-maze apparatus consisted of one long arm and two short arms. The end compartment, goal box, of each short arm contained either a social stimulus or a foraging stimulus. The test chicken was placed in the start box, and the test of 10 min began when the guillotine door of the start box was pulled via a lever system. White chalk lines divided the apparatus in sub-compartments that could be used for analysis. Illustrations: Yorrit van der Staay.

**Figure 3 animals-09-00053-f003:**
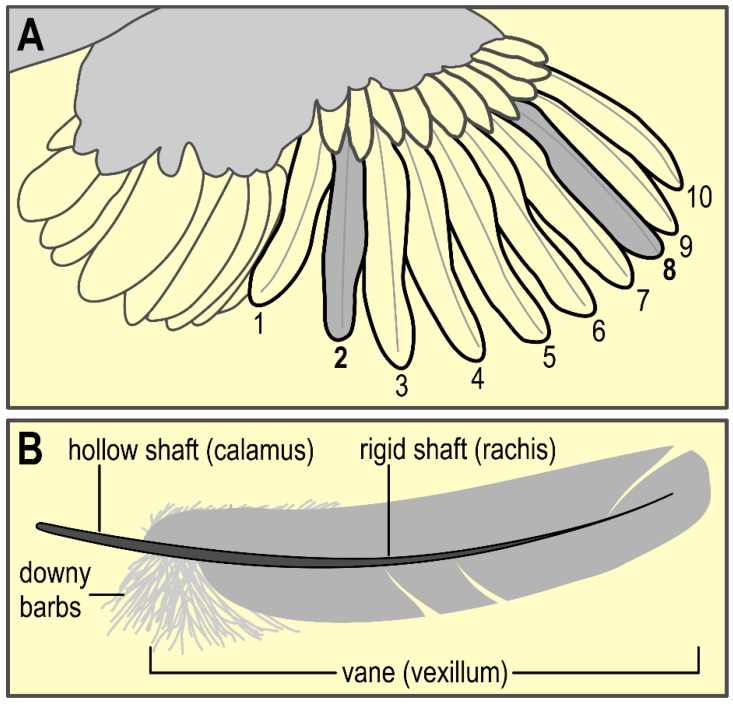
(**A**) Schematic representation of wing anatomy with numbering of primary feathers. For CORT analysis of the feathers, primary feather 2 and 8 were pulled in week 3 and 10. (**B**) Schematic representation of feather. Downy barbs, calamus, and rachis were removed prior to CORT analysis. Only the vanes were used for determination of corticosterone levels (illustration modified from [26]).

**Figure 4 animals-09-00053-f004:**
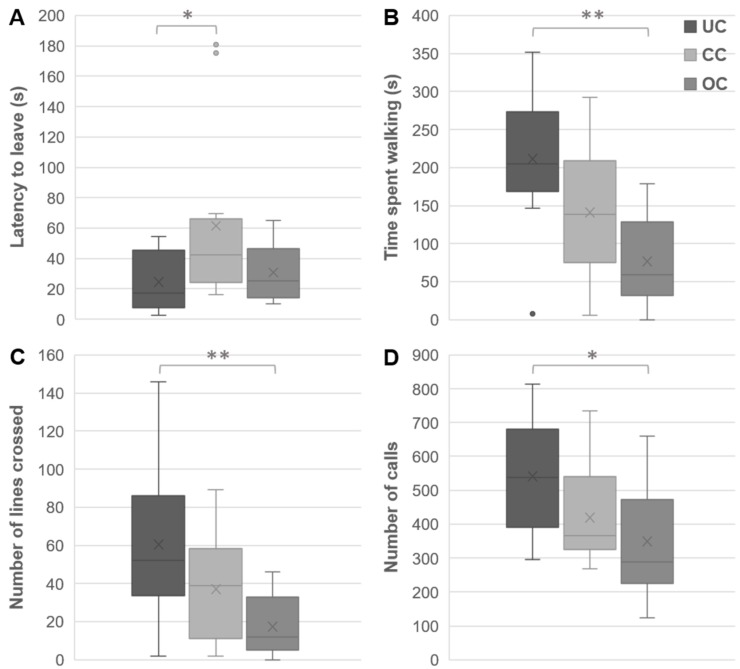
Effects of crowding condition on open field behavior. The latency to leave the start field (**A**), time spent walking (**B**), number of lines crossed (**C**), and number of distress calls emitted (**D**) during the OF test at three-weeks of age of chicks housed under undercrowding (UC), conventional crowding (CC), and overcrowding (OC) condition. The box ranges from the first to the third quartile, line indicates the median and cross indicates the mean, whiskers represent minimum and maximum without outliers, * *p* < 0.05, ** *p* < 0.01.

**Figure 5 animals-09-00053-f005:**
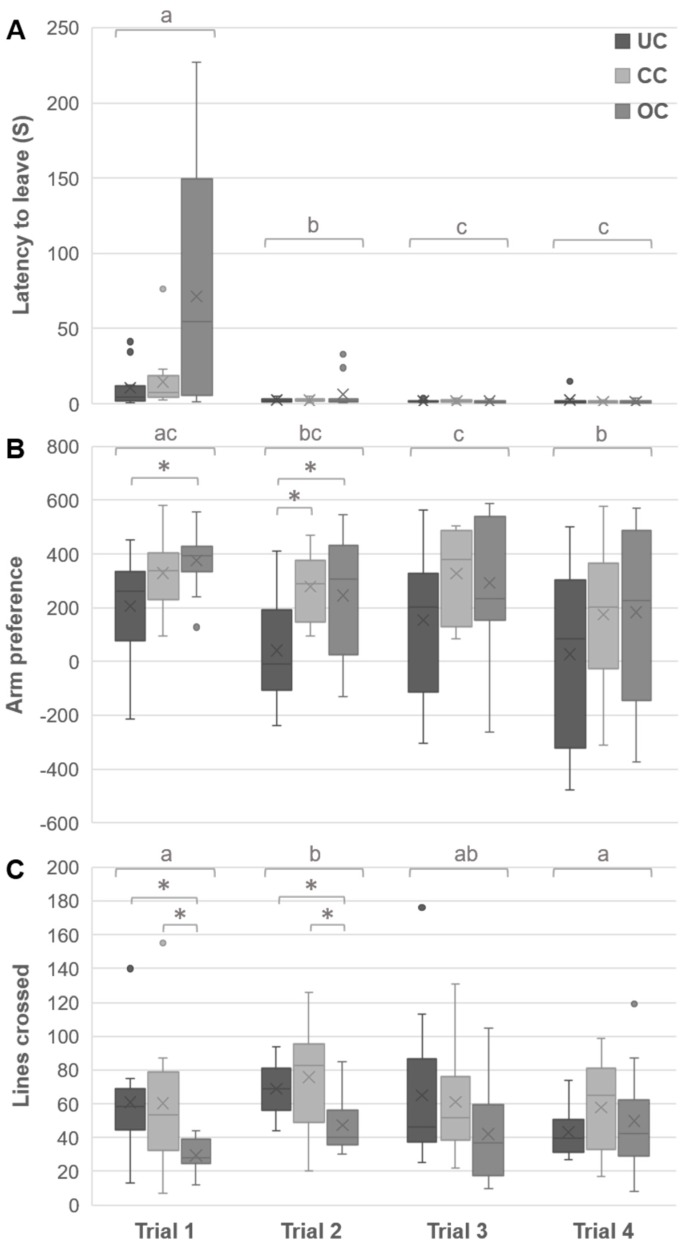
Effects of housing density on Y-maze behavior. The latency to leave the start-box (**A**), arm preference; with 0 indicating no preference, a positive value indicating the time spent more in the social arm, and a negative value indicating the time spent more in the foraging arm, in s (**B**), and number of lines crossed (**C**) of eight-week old chicks during the Y-maze test are depicted. Letters indicate trial differences across the groups, and asterisk indicate group differences within trials. The box ranges from the first to the third quartile, line indicates the median and cross indicates the mean, whiskers represent minimum and maximum without outliers, * *p* < 0.05 (UC = undercrowding, CC = conventional crowding, OC = overcrowding condition).

**Figure 6 animals-09-00053-f006:**
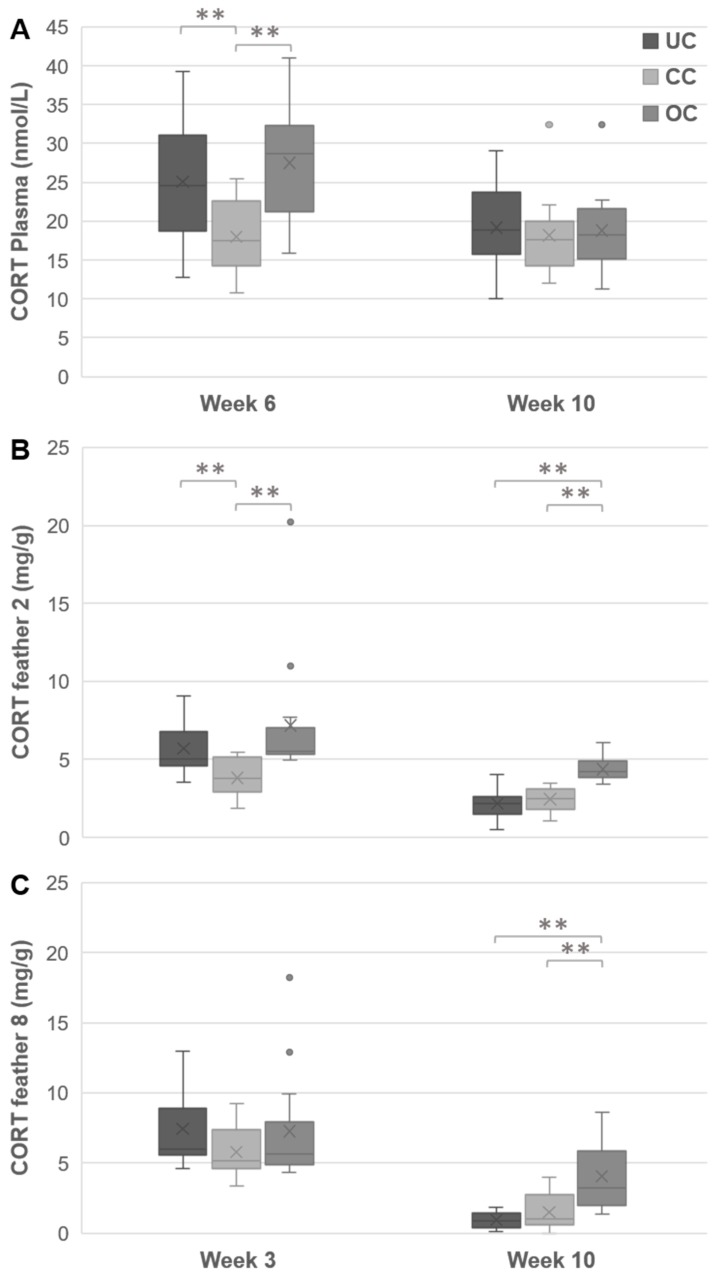
Effects of housing density on plasma CORT levels. The plasma CORT levels (nmol/L) in chicks aged 6 weeks and 10 weeks (**A**), CORT deposition (pg/mg) in feather 2 (**B**) and feather 8 (**C**) at 3 weeks and 10 weeks of age. The box ranges from the first to the third quartile, line indicates the median and cross indicates the mean, whiskers represent minimum and maximum without outliers, ** *p* < 0.01 (UC = undercrowding, CC = conventional crowding, OC = overcrowding condition).

**Table 1 animals-09-00053-t001:** Literature overview research on effects of stocking densities during the rearing period on the laying hen. Density is expressed in cm^2^ per chick. Per parameter is depicted if the study found an increase, decrease, or no difference (nd) with increasing stocking densities. If the study did not consider the specific parameter, it is designated with ‘-‘. The parameters considered are: body weight (BW), food intake (FI), uniformity of physique (U), fluctuating asymmetries (FA), mortality (M), plumage condition (PC), heterophil:lymphocyte (H:L) ratio, behavior, laying performance (LP).

Author	Year	Reference	Density (cm^2^/Chick)	BW	FI	U	FA	M	PC	H:L	Behavior	LP
Anderson & Adams	1992	[30]	192/221	(0–16) ↓ (16+) ↑	nd	nd	-	nd	nd	-	-	-
Bestman et al.	2009	[2]	(0–4) 294.1/476.2 (5–6) 416.7/555.6 (7–17) 952.4/010.1	-	-	-	-	-	-	-	↑ feather pecking	-
Blokhuis & van der Haar	1989	[31]	625/1250/3703.7	-	-	-	-	-	nd	-	↓ ground pecking ↑ feather pecking	-
Bozkurt et al.	2008	[12]	(0–4) 105.9/134.8/185.3 (4–16) 211.8/274.5/370.6	-	-	↑	-	-	-	↑	-	-
Bozkurt et al.	2006	[32]	(0–4) 105.9/134.8/185.3 (4–16) 211.8/274.5/370.6	↓	↓	-	-	nd	-	-	-	-
Carey	1986	[33]	1) 239/259/311 2) 222/259/311	↓	↓	-	-	-	-	-	-	↑ egg weight ↓ egg production decline
Hansen & Braastad	1994	[34]	769.2/1538.5	nd	nd	-	-	nd	↓	-	↓ ground pecking ↑ feather pecking	nd
Hester & Wilson	1986	[35]	344/516/1031	-	-	-	-	-	-	-	-	↓ hard-shelled:shell-less eggs ↓ eggs:hens
Huber-Eicher & Audige	1999	[36]	(0–2) > 285.7/< 285.7 (3–16) > 1000/< 1000	-	-	-	-	-	-	-	↑ feather pecking	-
Hunniford	2016	[37]	(0-5) 194.6/285.2 (6–16) 387.1/775 (16+) 690	-	-	-	-	-	-	-	↓ activity	-
Leeson & Summer	1984	[38]	293/586	nd	↓	-	-	-	-	-	-	↑ egg weight ↓ eggs
Moller et al.	1995	[39]	357.1/416.7/500	-	-	-	↑	-	-	-	↑ tonic immobility	-
Patterson & Siegel	1997	[40]	(0–6) 97.8/116.1/142.9/185.8 (6–16) 195.6/232.3/285.9/371.6	↓	(0–2) ↓ (2+) ↑	nd	-	nd	-	-	-	-
Pavan et al.	2005	[41]	(0–6) 210.5/228.6/250/275.9 (6–16) 357.1/416.7/500 (16+) 375/450/562.2	nd	nd	nd	-	-	-	-	-	nd
Wells	1972	[42]	700/930/1390/1860	nd	nd	nd	-	nd	↓	-	-	nd
Zepp et al.	2018	[43]	436.7/552.5	-	-	-	-	-	-	-	↑ feather pecking	-

**Table 2 animals-09-00053-t002:** Crowding conditions per group in available cm^2^ per chick. Age is depicted in weeks, and cages were enlarged at two time-points. Stocking densities were based on industry-numbers (taken as guiding value for conventional crowding) and a factor of three smaller or bigger to design the over- and undercrowding situation respectively.

Crowding Condition			
Age	Undercrowding	Conventional Crowding	Overcrowding
0–3	500 cm^2^	167 cm^2^	56 cm^2^
4–6	1000 cm^2^	333 cm^2^	111 cm^2^
7–10	1429 cm^2^	500 cm^2^	167 cm^2^

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
