# Peer review of "Stocking Density Affects Stress and Anxious Behavior in the Laying Hen Chick During Rearing"

_animals, 2019, doi:10.3390/ani9020053_

Round 1

Reviewer 1 Report

The paper is interesting and the topic is updated.

Only few remarks

Gnerally it could be better to underline why the OF was performed only one time but the YMaze test had 4 replicates. Consider che generally the results of these tests may be influenced by habituation to the apparatus.

Try to explain better also the value of arm preference. Is the arm preference In figure 5 expressed in seconds close to the social stimulus?

The page numbering is not correct in fact after table 1 the numbering restart from 1.

page 1 line 131 Table 2 not table 1

page 11 line 430 "..too large space..." instead of "...too large a space..."

Author Response

1. Gnerally it could be better to underline why the OF was performed only one time but the YMaze test had 4 replicates. Consider che generally the results of these tests may be influenced by habituation to the apparatus.

Response 1:

I would argue this is common procedure with the behavioural tests. The Open Field is primarily to test anxious behaviour by imposing a novelty on the chick, whereas the Y-maze is in the first trial to test the response to novelty, but over the trials intended to test sociality, which only becomes apparent after consecutive trails. Including consecutive trials in the OF would not inform us about additionally in any way.

2. Try to explain better also the value of arm preference. Is the arm preference In figure 5 expressed in seconds close to the social stimulus?

Response 2:

The reviewer makes a good point, and I have included an additional description below fig. 5, line 331-333, to clarify the values and have elaborated on the unit (in seconds)

Other remarks:

The page numbering is not correct in fact after table 1 the numbering restart from 1. --> adjusted

page 1 line 131 Table 2 not table 1 --> adjusted

page 11 line 430 "..too large space..." instead of "...too large a space..." --> that is also grammatically incorrect, therefore I changed the sentence entirely. (“..both more extreme housing densities…”) 

Reviewer 2 Report

Thank you for the opportunity to read your work. In general, this is an interesting study that makes a good attempt to investigate an important issue. It does need some editing with a fine-tooth comb to look for editing issues and consideration of some aspects of the statistical analysis, data presentation, and interpretation of results. I have provided specific comments below.

L14: There is an issue with using the term “stress hormones” – the hormones measured are activated by many things, stress being only one of them, and therefore using these terms can be misleading. The simple summary mentions the behavioural patterns at 6 weeks but not at 10 weeks, which makes the conclusion within the summary difficult to accept.

L13: ‘hen’ should be ‘hens’

L27: In the abstract, please provide the stocking density in the three crowding conditions

L31: Please provide an example of ‘anxious’ behaviour here

L44: Include the species latin name

L54: ‘Table 1’ should be in brackets

L60: Sentence starting ‘These factors’ should be reworded for clarity

Figure one isn’t that necessary. It would be enough to just put the rearing age range at line 52. If a similar figure was considered important, it would be better to have a timeline for the experiences of the chicken during the entire experiment including when each test was applied.

L66: While an animal’s perception is an important consideration for welfare, welfare can also be impaired even if an animal does not perceive a threat, stressor or negative state. It would be good to reference the five domains framework here as it outlines the importance of affective states in welfare, but does not neglect the other domains which are also important.

L71: Change ‘the’ to ‘an’

L77: Change ‘Up till now’ to ‘Until now’

L77: Some justification must be included of using CORT as a measure of chronic stress. It is recognised to be quite flawed in this regard and so it must be clearly outline why the method used has been validated as a chronic stress measure in this species, as opposed to its use as a measure of acute stress.

L81: Reference is needed for sentence starting “Anxiety can be defined as”

L82: Rated by whom? Is the text referring to these states in humans (e.g. self-rating) or in animals (e.g. rating by an observer)? Also, a reference is needed here.

L83: The statement that anxiety is reflected in fear and sociality needs more clarification.

L83: In reference to the sentence starting “Fear…” fear is an emotion and so cannot be directly measured. Fearfulness or fearful behaviour etc would be better here.

L85: Change ‘vice-versa’ to ‘Similarly’. Also, this sentence sounds odd because it is in the past tense unlike the sentences before and after.

L94: Reference is needed for sentence starting ‘Sociality can be’ and the sentence directly following it

From L102: There are quite a lot of methods details in this section, which should be in the subsequent section

L148: What was the temperature controlled at? (e.g. range)

L154: Industry in which country/region? Please also provide a reference.

L209: Figure titles should be underneath figures (Table titles go above)

L209: For the Y-maze in Figure two, it is not obvious that one arm is longer than the other two as per the description. Also, the stimuli should be included in the figure as that is an important part of the experimental protocol

L240: 5ml of? This sentence isn’t clear to me

L257: ‘Statistical’ is spelled incorrectly

L257: The stats analysis section needs more clarity and I would suggest focusing on each dependent variable at a time so it is clear how each was treated. For example, body weight is the first variable discussed in the results but it is not clear which of the stats tests were applied to it.

L268: Were the assumptions of normality rechecked after transformation? (also for L274)

L268: ‘Assessed’ is twice in this sentence

L270: What is meant by a ‘new’ variable in terms of the arm preference?

L277: ‘Consulted’ isn’t the right word here – ‘considered’ maybe?

L281: It is noted that the effect of crowding conditions was analysed using a MANOVA but which dependent variable is this for? Was the same analysis used for all of the variables of interest collected?

L285: The repeated measures aspect should be incorporated into the ANOVA that compares the effect of crowding condition on the CORT variables and the body weight, otherwise the assumptions of the ANOVA are violated

L292: It is stated that there was no effect of crowding on body weight but no details of the data or stats results are given. It is important that readers are presented with the data and results so they can make independent judgements, rather than rely solely on the author’s assessment.

L297: 3 decimal places are generally sufficient for the p values unless the first three decimals are all zeros

Figure 4 title: Be consistent with terms. “housing density” is used here while in the manuscript “crowding condition” is used

Figure 4: The figure looks like untransformed and non-backtransformed data. Make it clear in the figure titles if the analysis is actually based on transformed data.

L296 & 310: An effect on which behaviour? If the behaviours have been somehow combined for this analysis it is not made clear in the statistical analysis section. Also, the data must be presented in some way (either in the text or in a table/figure), not just the statistics

L311: Provide the statistics for the analyses mentioned in the sentence starting “There was no effect…”

Figure 5 is very busy. Try and reduce the amount of detail to make it easier to read. Even removing the background lines will help.

L319: A ‘marginal effect’ but the effect size was still in the high category according to the definition specified earlier in the manuscript?

L343: Be consistent as to whether Figure 5 or Fig. 5 style is used

- In section 3.5 the CORT results are discussed but figure 5 is referred to which is the behavioural results

- Figure 6 title has two full stops after ‘figure 6’

L356. ANOVAs does not need an ‘

L399. While this statement is true the Jones study specifically is measuring CORT activation in relation to a short-term stressor which is quite different to the current study which is looking at chronic conditions

L435: ‘Beneficiary’ should be ‘beneficial’

L452. Remove ‘for example’

- In the discussion there needs to be some discussion of how CORT is affected by other factors that are not related to stress (e.g. positive highly aroused states will also trigger increases in CORT) and how this impacts on the possible interpretation of the data from a welfare point of view, particularly for the UC group who will have more space for activity/locomotion etc

- The discussion must also outline why the data from the different trials is important and relevant for welfare. In the results, there is a lot made of the trial effects from the y-maze data but it isn’t clear how important this is. I also wonder how appropriate it was to analyse the different trials as individual categories rather than using a regression type analysis to get a single rate of change over the trials, which is what is mentioned as important in the introduction of the manuscript.

L35: I think that this statement: ‘and that this specific strain is capable to adapt to a three-fold in- or decrease” is overinterpreting the results and should be removed, especially given the limitations outlined by the authors in the discussion.

Author Response

Thank you for the opportunity to read your work. In general, this is an interesting study that makes a good attempt to investigate an important issue. It does need some editing with a fine-tooth comb to look for editing issues and consideration of some aspects of the statistical analysis, data presentation, and interpretation of results. I have provided specific comments below.

Response: thank you for your very thorough and helpful review. We hope to have addressed all the points sufficiently.

L14: There is an issue with using the term “stress hormones” – the hormones measured are activated by many things, stress being only one of them, and therefore using these terms can be misleading.

Response: the reviewer has a good point, replaced ‘Stress hormones’ with ‘corticosterone’, and included a small description of its involvement in the stress response.

L14: The simple summary mentions the behavioural patterns at 6 weeks but not at 10 weeks, which makes the conclusion within the summary difficult to accept.

Response: CORT measurements and behavioural tests did not occur at exactly the same time-points, which complicates communicating the results in a short and easy manner. We therefore made the decision to discuss all findings in a more general manner.

L13: ‘hen’ should be ‘hens’ --> replaced ‘hen’ with ‘hens’

L27: In the abstract, please provide the stocking density in the three crowding conditions --> added exact stocking densities

L31: Please provide an example of ‘anxious’ behaviour here

Response: we understand, and agree with, the importance to provide examples of these kind of abstract concepts, but we do not think the Abstract is a suitable place for this. An elaboration and definition of this behaviour is outlined in the main text, and we feel this is sufficient.

L44: Include the species latin name --> added Gallus gallus domesticus

L54: ‘Table 1’ should be in brackets --> put Table 1 in brackets

L60: Sentence starting ‘These factors’ should be reworded for clarity --> rephrased by putting part of the sentence as a subordinate clause at the end of the sentence

Figure one isn’t that necessary. It would be enough to just put the rearing age range at line 52. If a similar figure was considered important, it would be better to have a timeline for the experiences of the chicken during the entire experiment including when each test was applied.

Response: the reviewer makes a good proposal, and we have adjusted to figure to include experimental details.

L66: While an animal’s perception is an important consideration for welfare, welfare can also be impaired even if an animal does not perceive a threat, stressor or negative state. It would be good to reference the five domains framework here as it outlines the importance of affective states in welfare, but does not neglect the other domains which are also important.

Response: We are aware that welfare is a complex concept, and that there are multiple ways to define and assess it. The definition proposed is an adaptation to the five freedoms (the precursor of the five domains), and we therefore think it does already incorporate this important approach. In addition, we do not think it is necessary to outline all possible definitions here or go into a more detailed discussion of the different definitions. We do agree that particularly the phrasing is a bit confusing, so we have adjusted it.

L71: Change ‘the’ to ‘an’ --> check

L77: Change ‘Up till now’ to ‘Until now’ --> check

L77: Some justification must be included of using CORT as a measure of chronic stress. It is recognised to be quite flawed in this regard and so it must be clearly outline why the method used has been validated as a chronic stress measure in this species, as opposed to its use as a measure of acute stress.

Response: in this specific sentence we do not state that CORT is a measurement of chronic stress, we do state that corticosterone is a physiological expression/measurement of stress, but also list alternative physiological phenomena linked to CORT levels (metabolism, immune system, etc.). Next, we address that chronic high levels of CORT have been linked to unbeneficial physiological and behavioural consequences. To clarify we are not of the opinion CORT can solely be taken as indicative of chronic stress, we have tried to emphasize the importance of combining CORT measurements with other assessments (in this case behavioural parameters of fearfulness and sociality). See sentence in L84-86 “Since corticosterone has been implied in such a wide variety of physiological processes, its usefulness in the assessment of stress and welfare has been questioned [14]. It is therefore important to combine plain hormone levels with measures of behaviours indicative of stress.”

L81: Reference is needed for sentence starting “Anxiety can be defined as”

Response: the citation is included after the following sentence.

L82: Rated by whom? Is the text referring to these states in humans (e.g. self-rating) or in animals (e.g. rating by an observer)? Also, a reference is needed here.

Response: I remove the necessity of an actor by avoiding the word “rated”, instead I use “ranges” to still be able to address this property of anxiety.

L83: The statement that anxiety is reflected in fear and sociality needs more clarification.

Response: I would argue this is clarified in the sentences following (especially the last sentence of the paragraph): ‘Importantly, the intensity of the fearful response in the OF can be considered a measurement of overall anxiety, because the OF does not impose a direct threat of the chick.”

L83: In reference to the sentence starting “Fear…” fear is an emotion and so cannot be directly measured. Fearfulness or fearful behaviour etc would be better here. --> changed fear into Fearfulness

L85: Change ‘vice-versa’ to ‘Similarly’. Also, this sentence sounds odd because it is in the past tense unlike the sentences before and after. --> check, and changed the tense of the sentence

L94: Reference is needed for sentence starting ‘Sociality can be’ and the sentence directly following it

Response: I included references for the ‘sociality can be..’ sentence (Nordquist et al 2011, 2012), and the observation of the sentence following it is a personal observation, which I think has valid grounds since the Y-maze is a completely new environment to the chick, and would have a similar novelty-effect as the Open Field.

From L102: There are quite a lot of methods details in this section, which should be in the subsequent section

Response: this could be a preference difference, where personally I like to have a small summary of the main research question and methods at the end of the introduction and first paragraph of the discussion. This allows people who would prefer to scan the paper to just read the last paragraph of the intro and first paragraph of the discussion to get a good overview of the paper. I would prefer to leave it like this.

L148: What was the temperature controlled at? (e.g. range) --> added details (18-22 degrees Celsius)

L154: Industry in which country/region? Please also provide a reference. --> added country and reference to personal communication

L209: Figure titles should be underneath figures (Table titles go above) --> should be the case

L209: For the Y-maze in Figure two, it is not obvious that one arm is longer than the other two as per the description. Also, the stimuli should be included in the figure as that is an important part of the experimental protocol

Response: we have added the measurements to the behavioural tests, and stimuli location in text.  

L240: 5ml of? This sentence isn’t clear to me --> I understand the confusion and have hopefully clarified with: For the actual extraction, a basis of 5 ml 80% methanol was added, plus 1 ml 80% methanol per 0.03 gram flakes, followed by…

L257: ‘Statistical’ is spelled incorrectly --> corrected

L257: The stats analysis section needs more clarity and I would suggest focusing on each dependent variable at a time so it is clear how each was treated. For example, body weight is the first variable discussed in the results but it is not clear which of the stats tests were applied to it.

Response: we definitely understand the reviewers point, and have included the specifics on Body Weight analysis (which was indeed forgotten). We decided to add clarity by specifying the different variables after each mentioned test, but found it more clear to keep the structure are before (since some tests apply to different variables, and in this way repetition is avoided).

L268: Were the assumptions of normality rechecked after transformation? (also for L274)

Response: yes, and a sentence has been added where we clarify that the assumption of normality was met after transformation.

L268: ‘Assessed’ is twice in this sentence --> removed one ‘Assessed’

L270: What is meant by a ‘new’ variable in terms of the arm preference? --> replaced ‘new’ with ‘compounded’

L277: ‘Consulted’ isn’t the right word here – ‘considered’ maybe? --> ‘consulted’ replaced by ‘considered’

L281: It is noted that the effect of crowding conditions was analysed using a MANOVA but which dependent variable is this for? Was the same analysis used for all of the variables of interest collected?

Response: yes, and we hope that we have clarified this now in the text by listing the specific variables subjected to which tests

L285: The repeated measures aspect should be incorporated into the ANOVA that compares the effect of crowding condition on the CORT variables and the body weight, otherwise the assumptions of the ANOVA are violated

Response: we have done this, and understand that our description of statistics did not clarify this before. We hope we have adjusted the text now sufficiently that this comes across.

L292: It is stated that there was no effect of crowding on body weight but no details of the data or stats results are given. It is important that readers are presented with the data and results so they can make independent judgements, rather than rely solely on the author’s assessment. --> good point, statistics have been added

L297: 3 decimal places are generally sufficient for the p values unless the first three decimals are all zeros --> changed statistics to three decimals

Figure 4 title: Be consistent with terms. “housing density” is used here while in the manuscript “crowding condition” is used --> replaced ‘Housing density’ with ‘crowding condition’

Figure 4: The figure looks like untransformed and non-backtransformed data. Make it clear in the figure titles if the analysis is actually based on transformed data. --> the figures show the untransformed data

L296 & 310: An effect on which behaviour? If the behaviours have been somehow combined for this analysis it is not made clear in the statistical analysis section. Also, the data must be presented in some way (either in the text or in a table/figure), not just the statistics

Response: before looking into the effects of the separate behaviours, we evaluated the statistics of the effects of crowding on all behaviours in the Open Field. We believe this is necessary before looking at the separate analysis. However, we think that the separate behaviours are much more interested than the overall analysis therefore we decided not to include a figure or table. If the reviewer disagrees we could consider leaving out these statistics completely.

L311: Provide the statistics for the analyses mentioned in the sentence starting “There was no effect…” --> statistics added

Figure 5 is very busy. Try and reduce the amount of detail to make it easier to read. Even removing the background lines will help.

Response: we understand the figure is complex, but still find it the best way to depict our findings. Also, we think the lines help for assessing the exact values. This might be a personal preference difference, and if it is truly obstructing we will re-consider.

L319: A ‘marginal effect’ but the effect size was still in the high category according to the definition specified earlier in the manuscript?

Response: I understand the confusion here, the effect size is indeed large, but the significance value is 0.051. I know that usually this means NOT SIGNIFICANT, but seeing that the effect size is large I leave the statistics in but point out the 0.051 and large effect size combination.

L343: Be consistent as to whether Figure 5 or Fig. 5 style is used

- In section 3.5 the CORT results are discussed but figure 5 is referred to which is the behavioural results --> changed all fig. 5 into Figure 6

- Figure 6 title has two full stops after ‘figure 6’ --> removed one stop

L356. ANOVAs does not need an ‘ --> removed ‘

L399. While this statement is true the Jones study specifically is measuring CORT activation in relation to a short-term stressor which is quite different to the current study which is looking at chronic conditions

Response: the current study is looking both at short-term levels (basal CORT in plasma) and chronic levels (which is a build-up of plasma CORT levels), therefore we do think the Jones study is relevant for interpreting our data.

L435: ‘Beneficiary’ should be ‘beneficial’ --> changed ‘beneficiary’ into ‘beneficial’

L452. Remove ‘for example’ --> removed ‘for example’

- In the discussion there needs to be some discussion of how CORT is affected by other factors that are not related to stress (e.g. positive highly aroused states will also trigger increases in CORT) and how this impacts on the possible interpretation of the data from a welfare point of view, particularly for the UC group who will have more space for activity/locomotion etc

Response: this is an interesting point, and we have included a paragraph in the discussion section where we address the increased CORT values as a consequence to increased physical activity, see L434-443

- The discussion must also outline why the data from the different trials is important and relevant for welfare. In the results, there is a lot made of the trial effects from the y-maze data but it isn’t clear how important this is

Response: We have some difficulties understanding what Reviewer 2 exactly means here.

...I also wonder how appropriate it was to analyse the different trials as individual categories rather than using a regression type analysis to get a single rate of change over the trials, which is what is mentioned as important in the introduction of the manuscript.

Response: repeated measurements were analysed using a repeated measure analysis (PROC GLM REPEATED). A regression type analysis is less sensitive for fluctuations in measurements on different time-points.

L35: I think that this statement: ‘and that this specific strain is capable to adapt to a three-fold in- or decrease” is overinterpreting the results and should be removed, especially given the limitations outlined by the authors in the discussion.

Response: we agree with the referee and have rephrased it to ‘… within the adaptive capacity of the chick’, which is a less strong statement but does address the change in CORT levels over time.